Comparative cephalopod shell strength and the role of septum morphology on stress distribution

Lemanis Robert Robert.Lemanis@rub.de 1
Zachow Stefan 2
Hoffmann René 1
1 Institute of Geology, Mineralogy, and Geophysics, Ruhr-Universität Bochum , Bochum , Germany
2 Department of Visual Data Analysis, Zuse Institute Berlin , Berlin , Germany
Weiss Ingrid
Electronic publication date: 2016 Sep 13
Publication date: 2016
Volume: 4
Electronic Location ID: e2434
Received 2016 Apr 7; Accepted 2016 Aug 12
Copyright: ©2016 Lemanis et al.
Copyright year: 2016
Copyright holder: Lemanis et al.
License: This is an open access article distributed under the terms of the Creative Commons Attribution License, which permits unrestricted use, distribution, reproduction and adaptation in any medium and for any purpose provided that it is properly attributed. For attribution, the original author(s), title, publication source (PeerJ) and either DOI or URL of the article must be cited.
License URL: https://creativecommons.org/licenses/by/4.0/

Keywords: Finite element analysis, Biomechanics, Septal complexity, Ammonoidea, Nautilida, Computed tomography

Funding: Deutsche Forschungsgemeinschaft HO 4674/2-1 This work is funded by a grant from the Deutsche Forschungsgemeinschaft, grant: HO 4674/2-1. The funders had no role in study design, data collection and analysis, decision to publish, or preparation of the manuscript.

==============================
The evolution of complexly folded septa in ammonoids has long been a controversial topic. Explanations of the function of these folded septa can be divided into physiological and mechanical hypotheses with the mechanical functions tending to find widespread support. The complexity of the cephalopod shell has made it difficult to directly test the mechanical properties of these structures without oversimplification of the septal morphology or extraction of a small sub-domain. However, the power of modern finite element analysis now permits direct testing of mechanical hypothesis on complete, empirical models of the shells taken from computed tomographic data. Here we compare, for the first time using empirical models, the capability of the shells of extant Nautilus pompilius, Spirula spirula, and the extinct ammonite Cadoceras sp. to withstand hydrostatic pressure and point loads. Results show hydrostatic pressure imparts highest stress on the final septum with the rest of the shell showing minimal compression. S. spirula shows the lowest stress under hydrostatic pressure while N. pompilius shows the highest stress. Cadoceras sp. shows the development of high stress along the attachment of the septal saddles with the shell wall. Stress due to point loads decreases when the point force is directed along the suture as opposed to the unsupported chamber wall. Cadoceras sp. shows the greatest decrease in stress between the point loads compared to all other models. Greater amplitude of septal flutes corresponds with greater stress due to hydrostatic pressure; however, greater amplitude decreases the stress magnitude of point loads directed along the suture. In our models, sutural complexity does not predict greater resistance to hydrostatic pressure but it does seem to increase resistance to point loads, such as would be from predators. This result permits discussion of palaeoecological reconstructions on the basis of septal morphology. We further suggest that the ratio used to characterize septal morphology in the septal strength index and in calculations of tensile strength of nacre are likely insufficient. A better understanding of the material properties of cephalopod nacre may allow the estimation of maximum depth limits of shelled cephalopods through finite element analysis.

Introduction

The cephalopod shell is a complex structure that has multiple functions including buoyancy regulation and protection of the animal against predations, water pressure, and strong currents (Derham, 1726; Hewitt & Westermann, 1997; Hassan et al., 2002). These shells, whether they be the internal shells of Recent coleoids—such as Spirula and Sepia—or the external shells of Nautilus and the extinct ammonoids, must resist the hydrostatic pressure of the surrounding water, which depends on the mechanical strength of the shell. This is due to the fact that the internal pressure of the chambers of cephalopod shells is only around one atmosphere (Swammerdam, 1758; Buckland, 1836; Denton & Gilpin-Brown, 1961; Denton & Gilpin-Brown, 1966). The maximum depth of some extant, shelled cephalopods ranges from several hundred meters to around 1,000 m (Bruun, 1943; Dunstan, Ward & Marshall, 2011; Hoffmann & Warnke, 2014); depth estimates for ammonoids and extinct nautiloids cover a similar range and, in some cases, even deeper (Westermann, 1996; Westermann, 1999). Water pressure at 1,000 m depth is around 100 times atmospheric pressure at sea level. Accordingly, the shell must be able to resist this pressure while also resisting deformation that would impact the buoyancy ability of the shell. Buckland (1836) hypothesized that the extreme folding of the septa—mineralized partitions that divide the shell into a series of chambers—famously displayed by ammonoids, was an adaptation to increase the resistance of the shell against implosion by hydrostatic pressure. This idea has become known as the Buckland model.

Alternatively, Pfaff (1911) postulated that the increase in septal complexity buttressed the final septum against the hydrostatic pressure acting against its face as pressure would be transmitted through the soft-tissue (Hewitt & Westermann, 1986). This idea is known as the Pfaff model. Both models were later complemented by the Westermann model that argued septa support previous whorls through management of bending moments (Westermann, 1958; Hewitt & Westermann, 1986). In this view the septa are compared to springs, and septal complexity helped buttress the inner whorls against indirect hydrostatic pressure transmitted through the outermost whorl (Hewitt & Westermann, 1987a; Hewitt & Westermann, 1997).

The mechanical explanation of septal complexity has been heavily debated (e.g., Chamberlain & Chamberlain, 1985; Westermann, 1985; Saunders, 1995; Hewitt & Westermann, 1997). Alternative hypotheses for the function of ammonitic septa focus on their potential physiological functions (e.g., Daniel et al., 1997; Perez-Claros, 2005; Perez-Claros, Oloriz & Palmqvist, 2007; Lemanis et al., 2016). For a comprehensive review of potential functions of complex septa see Klug & Hoffmann (2015).

One difficulty of testing mechanical hypotheses is accurately modelling the complex geometry of the shells and septa in order to test them. Generally, research in this field focuses on a small sub-sample of the domain of interest in order to model the complexity of the morphology or simplifying a larger domain of interest (Westermann, 1973; Jacobs, 1990; Hewitt, 1996; Daniel et al., 1997; Hassan et al., 2002; De Blasio, 2008). Finite element analysis (FEA) is one of the most promising, and powerful techniques to test the mechanical properties of the shell and septa.

FEA has become a popular method in palaeontology to address biomechanical questions (Witzel & Preuschoft, 2005; Rayfield, 2006; Rayfield, 2007; Rayfield, 2009; Falkingham, Margetts & Manning, 2010; Tseng, 2013; Lautenschlager, 2014; Button, Rayfield & Barrett, 2014; Cox, Rinderknecht & Blanco, 2015; Ledogar et al., 2016). This technique converts a continuous structure into a set of discrete elements, in 3D these are typically tetrahedra or cubes, over which a set of solutions to a given problem are solved and then compiled to a representative continuum solution (Turner et al., 1956; Cook et al., 2001). One of the first applications of FEA in zoology was Guillet, Doyle & Rüther (1985) who studied the beak of the African shoebill Balaeniceps rex; studies using FEA on ammonites quickly followed Hewitt & Westermann (1987b).

Following this, Hewitt published several FEA studies focusing on nautiloids (Hewitt & Westermann, 1987b; Hewitt et al., 1989; Hewitt et al., 1993). Hewitt et al. (1989) modeled the final septum and surrounding shell region of the Carboniferous nautiloid Michelinoceras unicamera and subjected it to a simulated, external pressure load to estimate implosion depth (1,125 m). The septal neck region of M. unicamera was modeled by Hewitt et al. (1993) who analyzed the relative strength of the nacreous/chitinous complex of the septal neck and connecting ring. The two most recent applications of FEA (Daniel et al., 1997; Hassan et al., 2002) on cephalopods focused on the previously discussed mechanical hypothesis of septal complexity and came to contradictory conclusions.

Both Daniel et al. (1997) and Hassan et al. (2002) recreated approximate septal morphologies using sequential Fourier equations and place the simulated structures in a cylindrical shell model. Both the shell and the septa have uniform thickness. Daniel et al. (1997) concluded that as septal complexity increases, stress on the septum also increases; furthermore as folding increases, stress is focused towards the center of the septum. Hassan et al. (2002) modeled three hypothetical morphologies of increasing complexity going from their Goniatite model to Ammonite “A” and Ammonite “B”. Their analysis showed a decrease of maximum principal stress and shear stress in the septum as complexity increased, challenging the results of Daniel et al. (1997), though both agreed that a simple, semi-hemispherical septum would be more resistant to stress than folded septa. Both studies used simplified, theoretical septal morphologies that focused on a single septum with uniform thickness exposed to a pressure force of 0.1 MPa. These simplifications likely limit the comparability of these models to true septa.

Here we present the first FEA of empirical models of cephalopod shells. The whole shells of Nautilus pompilius, Spirula spirula, and Cadoceras sp. are modeled here (Fig. 1) to test the pattern of stress development and distribution across the shell and compare stress development and distribution in the septa due to pressure and point loading. The use of empirical models of shells allows us to bypass the geometric problems of Daniel et al. (1997) and Hassan et al. (2002) and gives a more accurate representation of stress due to realistic loading conditions.

Figure 1 Shell renderings.

Volume renderings of the three specimens used in this study (A, C, E). Volume renderings cut along a median section to show the internal structure of the shell and cross-sectional septal shape (B, D, F).

Material & Methods

Specimens and segmentation

A total of three specimens and four models were used in this study (Table 1). One shell of Nautilus pompilius, collected from the Philippines, was scanned at the Steinmann Institute at the University of Bonn with a phoenix v|tome|x s (General Electric) and an isotropic voxel size of 175 µm (the same specimen used in Hoffmann et al. (2014)). One shell of Spirula spirula from Thailand was scanned using a phoenix nanotom m (General Electric) at the TPW Prüfzentrum (Neuss, Germany) with an isotropic voxel size of 9 µm. Cadoceras sp. was scanned at the Advanced Photon Source at Argonne National Labs using phase contrast synchrotron tomography with an isotropic voxel size of 0.74 µm. S. spirula represents the simplest septal morphology, possessing domed shaped septa. N. pompilius possess synclastic septa, meaning they are adorally concave, whereas Cadoceras sp. possess anticlastic septa (adorally convex) typical of ammonites (Fig. 1). According to the observed terminal changes in shell morphology N. pompilius and S. spirula are adult specimens (Seilacher & Gunji, 1993). Only the ammonitella (hatchling shell), indicated by the nepionic constriction, of Cadoceras sp. is preserved and segmented.

Table 1 General information on specimens, finite element models, and boundary conditions.

Specimen	Age	Locality	Total surface area (mm2)	Elements	Nodes	Force applied by 8 MPa pressure (N)	Scaled point force (N)	
Cadoceras sp. (5 septa)	Callovian	Russia	6.70	1,780,464	428,535	32.22	10	
Cadoceras sp. (10 septa)	“	“	8.36	2,373,913	517,318	21.41	12.48	
Spirula spirula	Recent	Thailand	2720.99	2,058,092	583,565	7262.31	4063.45	
Nautilus pompilius	Recent	Phillipines	152400.35	2,434,318	723,832	693270.16	227590.56	

Segmentation of all specimens began with the application of an initial, global threshold that was manually refined using ZIBAmira (Zuse Institute, Berlin). For further details on the reconstruction of the Cadoceras sp. see Lemanis et al. (2015). Two Cadoceras sp. models were generated for this study. The first being the ammonitella with ten septa (the maximum number of preserved septa in the ammonitella) and a separate model with five septa. These two models offer a comparative demonstration of how the amplitude of septal folds affects stress distribution. The septa of S. spirula and N. pompilius only show minor changes through ontogeny and none of these changes are used to reconstruct the animal’s ecology.

Meshing and finite element modelling

The final labelfields were used as the basis for the construction of a series of stereolithographic surface meshes (stl). Due to the high resolution of the original scans, the resulting stl files were oversampled, a single stl was composed of over 15 million faces. In order to produce workable sized meshes, all data sets were down-sampled to twice the original voxel size. Testing was performed to measure the effects of the down-sampling on the resulting finite element analysis. Similar to the work of McCurry, Evans & McHenry (2015), the original data sets of N. pompilius and S. spirula were down-sampled to 2×, 4×, and 6× of their original voxel size. The resulting FE-models were then compared against an edited stl from the original data set. FEA was performed with these models under the pressure loading described later in this section. In both cases, the results of the edited stl and the 2× down-sampled mesh showed nearly identical results, within 10% of the peak max. principal stress on the final septum. Therefore, 2× down-sampled datasets were used as producing a final mesh from these data were quicker than editing the original mesh and produced an initial mesh with less errors. The 4× and 6× down-sampled meshes produced results both outside of the 10% limit and showed visible degradation of fine-scale features, e.g., the ridges on the external surface of the S. spirula shell. Furthermore, the 6× down-sampled mesh appeared rough in small areas of high curvature, such as the internal surface of the septal necks.

Surface mesh editing and volumetric meshing were performed in ZIBAmira. All stl files were re-meshed using the implementation of the algorithm of Zilske, Lamecker & Zachow (2008). Surface meshes were edited to correct for triangle aspect ratio, dihedral angles, and tetrahedra quality. Volumetric meshes composed of first-order tetrahedral elements (TET4) were generated via the advancing front method (e.g., Löhner & Parikh, 1988).

Convergence testing is a vital step in FEA that is necessary to determine the resolution of the meshes necessary for the analysis to solve completely and show an accurate stress distribution across the structure of interest (Bright & Rayfield, 2011; Walmsley et al., 2013). The convergence models were formed by altering the mesh size during surface generation and models were considered converged both upon visual inspections, when no further development of stress in the structure could be detected, and when the maximum values of stress in the models were within 10%. Final volumetric mesh sizes are in Table 1. Both four-node tetrahedral elements (TET4) and 10 node elements (TET10) were tested although the TET10 models proved to be too large to solve using the available computation resources (Windows PC with an Intel Xeon E5-1620 CPU @3.6 GHz, 65 GB of RAM, Nvidia Quadro 6000 with 4 GB of video RAM).

Finite element analysis

All FEA were performed with Mecway v4.0 (Mecway Limited, New Zealand). For all specimens, models were run as a static 3D, linear analysis with the isotropic material properties of mollusk nacre: a Young’s modulus of 50 GPa and a Poisson’s ratio of 0.29 (Hewitt, 1996; Daniel et al., 1997; Hassan et al., 2002). It should be noted that nacre tablets have orthotropic properties; however these properties are poorly studied for cephalopods as most studies focus on gastropods and bivalves (Jackson, Vincent & Turner, 1988; Barthelat et al., 2006; Bertoldi, Bigoni & Drugan, 2008). Due to the poorly constrained material properties of cephalopod nacre used here, the results should only be interpreted comparatively between the specimens. Loading cases are divided between pressure loads and point force loads.

Pressure loading

Simulation of hydrostatic pressure is done by applying a pressure load, i.e., a load normal to the face of an element, to every external face of the model. This includes faces of the exterior shell wall, the interior surface of the body chamber and the external face of the final septum. All specimens are subjected to an 8 MPa pressure, roughly equivalent to 785 m depth. N. pompilius is further testing at 2 MPa, 4 MPa, and 6 MPa to model the development of stress in the final septum. No single constraint is present in any pressure load case. It is worth noting that this is a simplified hydrostatic load as pressure is modeled here as a uniform field while true hydrostatic pressure would increase along the length of the shell with depth; pressure would be higher on the bottom of the shell than the top. Furthermore, the minor pressure exerted by the gas within the chambers (<1 atm/0.1 MPa) is ignored (Denton & Gilpin-Brown, 1966).

Point loading

Additional point forces are modeled to test the resistance of the shell to simulated bite forces. For all specimens, two situations are modeled. The first situation has a point load on the chamber wall, mid-way between the two septa. The second situation is a point load applied along the suture line, the attachment of the septa to the shell wall. In both cases the point force is applied along the side of the shell while the opposite side is constrained against translation and rotation in three dimensions. Point forces are applied to a single node with a resultant force of 10 N for Cadoceras sp. with five septa. Point forces are scaled (Table 1) to the total surface area of each specimen (Dumont, Grosse & Slater, 2009).

Septal strength index

Septal strength index (SSI; Eq. (1)), which calculates the supposed implosion depth of a shell based on septal curvature and thickness, calculations were done on median sections of the final septa of N. pompilius and S. spirula following the work of Westermann (1973), and transverse sections of the final septa of N. pompilius (Westermann, 1985). (1) δR∗1000

δ is thickness and R is the radius of curvature. Minimum septal thickness and radii of curvature are measured in ImageJ (Schneider, Rasband & Eliceiri, 2012). Radius of curvature is computed using the “ThreePointCircularROI” plugin (G. Landini, http://www.mecourse.com/landinig/software/software.html). Approximate depth limits are calculated using a conversion factor derived from Westermann (1973).

Results

Pressure

All tested shells show the same general pattern of stress distribution: overall the shell shows mostly compression (negative max. principal stress) or very minor tension (positive max. principal stress) while the final septum is under the highest tension as illustrated by elevated max. principal stress (green to black areas of the shell, Fig. 2). All septa show high tension along the area where the septa are attached to the shell wall, while the final septum of N. pompilius shows the most widespread distribution of high max. principal stress along the septal face compared to all other specimens (Fig. 3A). Only N. pompilius and Cadoceras sp. show notable stress on the external shell wall, generally corresponding to the area of the suture. Cadoceras sp. with five septa is the only model to show a global average compression while N. pompilius shows the highest global average values (Table 2). All tested shells also show a development of high tension around the siphuncular foramen (the only discontinuity in the septal surface).

Figure 2 Stress due to hydrostatic pressure.

Results of the finite element analyses of N. pompilius (A1–A3), S. spirula (B1–B3), Cadoceras sp (C1–C3, D1–D3). Each model is loaded with a pressure load of 8 MPa. Forces are directed normal to the exterior surface of the model, comprising the external shell wall, internal faces of the body chamber, and the external (adapertural) faces of the final septum. Nautilus nacre has a theoretical tensile strength of about 130 MPa (Westermann & Ward, 1980) which is used here as the maximum value of the provided scale. Positive max. principal stress indicates areas of tension while negative max. principal stress indicates areas of compression. Areas outside the range of the scale are rendered in black. Black areas on the septum are above the 130 MPa maximum. Using the 130 MPa strength estimate, black areas are the areas most likely to fail under the modelled pressure.

Figure 3 Comparative stress due to hydrostatic loading and development of stress in Nautilus pompilius under increasing pressure.

(A) Measurements of max. principal stress taken along a horizontal transect through the middle of the final septum. Note the asymmetric distribution of max. principal stress in the septum, likely due to the asymmetry of the shell itself. (B) Plot of the max. principal stress taken from the nodes of N. pompilius under four different (2, 4, 6, and 8 MPa) pressure loads. Peaks in this data correspond to the elevated max. principal stress that develops on the final septum. The bilateral symmetry of the graph is due to the bilateral symmetry of the shell itself. Note that as the hydrostatic pressure increases, certain regions increase in max. principal stress at a faster rate than other regions.

Table 2 Selected values of maximum principal stress under all loading conditions.

Maximum/minimum in the center and ventral margins are measured in median sections along a line connecting two nodes on the adoral and adapical septum surfaces. These values should be interpreted comparatively.

	Cadoceras sp. 10 Septa	Cadoceras sp. 5 Septa	Spirula spirula	Nautilus pompilius	
8 MPa pressure	
Average max principal stress (MPa)	1.4	−1.09	2.45	4.25	
Peak max. principal stress on final septum (MPa)	332.18	293.14	154	272.2	
Max/Min stress in the center of the final septum (MPa)	21.92/10.81	22.10/−4.02	88.54/71.45	150.21/97.10	
Max/Min stress in the ventral margin of the final septum (MPa)	119.10/46.97	183.75/77.57	110.27/0.36	170.10/32.62	
Point force on chamber wall	
Average maximum principal stress (MPa)	687	584.30	474.03	254.58	
Peak max. principal stress on internal shell surface (MPa)	18920.21	17983.78	47,071	79987.09	
Point force on suture	
Average maximum principal stress (MPa)	735.44	552.22	416.89	457.55	
Max. maximum principal stress on internal shell surface (MPa)	7750.25	10468.20	28,136	73318.09	

The ontogenetic development of the septa in Cadoceras sp. demonstrates an increase both in peak max. principal stress and the development of high max. principal stress as the sutural amplitude increases. There are no new lobes or saddles developed within the ontogenetic window between the 5th and 10th septa, though there is a slight dorso-ventral elongation of the septal surface. The adoral most edge of the saddles are the weakest points of the shell of Cadoceras sp., compare this with the development of high tension along most of the suture of S. spirula and N. pompilius though there is an asymmetry of stress development in these models.

Progressive loading of N. pompilius under increasing pressure shows the development of high tension along the suture and the lateral edges of the siphuncular foramen (Figs. 3B and 4). Furthermore, max. principal stress is diverted from the lateral regions of the septal surface corresponding to the area where the septum curves into the shell wall. Assuming the values of tension from the FEA are accurate (an incorrect assumption but one used here for illustrative purposes), the depth of implosion of this shell is between 4 and 6 MPa (roughly equivalent to 390–590 m below sea level). The calculated SSI (Westermann, 1973) estimates depth ranges from 406 m to 1,304 m (Table 3).

Figure 4 FEA results in Nautilus pompilius under increasing hydrostatic pressure.

Development of max. principal stress on the final septum of N. pompilius under increasing (2 (A), 4 (B), 6 (C), and 8 (D) MPa) hydrostatic pressure. The results of these models are graphically illustrated in Fig. 2B. The region of septal attachment to the shell wall and the regions immediately lateral to the siphuncular foramen are the weakest regions in this model.

Table 3 Septal strength index (SSI) and depth approximations.

Radius of curvature	Septal strength index	Depth approximation (m)	
Nautilus pompilius	
Median section			
18.39	43.50	1304.92	
23.87	33.51	1005.27	
25.05	31.93	957.97	
34.01	23.52	705.61	
25.76	31.06	931.66	
23.40	34.18	1025.53	
28.53	28.04	841.19	
22.52	35.53	1065.80	
21.87	36.58	1097.40	
28.65	27.92	837.73	
Transverse section			
26.88	29.76	892.91	
59.11	13.53	406	
Spirula spirula	
Median section			
3.14	52.83	1584.94	
3.36	49.34	1480.13	
3.78	43.88	1316.55	
3.55	46.72	1401.49	
3.26	50.97	1529.16	

Point force

Point forces oriented along the suture show lower maximum stress compared to point forces on the chamber wall for all specimens (Table 2). Cadoceras sp. with 10 septa shows the largest decrease in max. principal stress between the chamber point force and the suture point force with a 59% decrease while N. pompilius shows the lowest decrease at 8%. Both Cadoceras sp. with five septa and S. spirula show similar decreases at 42% and 40% respectively.

The point force along the suture line of both Cadoceras sp. models are the only results that do not exceed the applied universal threshold (Fig. 5).

Cadoceras sp. shows a marked decrease in global average max. principal stress with increasing numbers of septa (Table 2). The 10 septa model shows a 15% decrease in global average max. principal stress under the chamber point force load while the point force suture load shows a 25% decrease in global average max. principal stress.

Figure 5 Stress due to point force loading.

Point force loads were directed along the suture (A–D) and the unsupported chamber wall (E–H). Black regions are areas where the max. principal stress exceeds the limits of the scale. The magnitude of stress due to the point load decreases when the load in directed along the suture line rather than the chamber wall. Cadoceras sp. shows the greatest decrease in max. principal stress when the point load is moved to the suture line. (A, E) Nautilus pompilius. (B, F) Spirula spirula. (C, G) Cadoceras sp. with ten septa. (D, H) Cadoceras sp. with 5 septa.

Discussion

Models and resolution

The resolution of the initial scan, reported here as voxel size, is an important parameter as it directly controls the accuracy of the scanned morphology that forms the basis for the construction of the finite element models. As a general rule, the resolution of the scan should be at least half that of the structure of interest, i.e., a shell with a thickness of 1 mm should have a voxel size of at least 0.5 mm. Specimens used here were scanned with the highest available resolution, for example the minimum thickness of the final septum of Cadoceras sp. is 3.27 µm while the voxel size is 0.74 µm. The effects of insufficient resolution was seen during the resampling of the datasets as a high degree of resampling (4× and 6×) showed visible distortion of fine scale features and made areas of high curvature appear rougher.

The forces applied to each model were scaled to the surface area of that model. This was done in order to eliminate the effects of size, as each shell is of considerably different diameter (Fig. 1). The difference in size is due to both differences in adult size between species and differences in the ontogenetic stage: N. pompilius and S. spirula are both adults while Cadoceras sp. is a hatchling. Point forces were manually scaled while the pressure loads were automatically scaled as pressure exerts a force relative to the surface area over which that pressure is applied. Differences between the models are therefore due to shape, which allows us to directly compare the effects of shell and septal shape on the stress developed on the shell and the final septum. This allowed us to test the Buckland and Pfaff models but the Westermann model can’t be addressed with the data presented here because the Cadoceras sp. possessed only one complete whorl.

Pressure and comparisons with previous results

Both Daniel et al. (1997) and Hassan et al. (2002) agreed that spherical shaped septa would show less stress due to hydrostatic pressure when compared to more complexly folded septa. Our results corroborate this as S. spirula shows the lowest max. principal stress over the septum face as well as the lowest peak max. principal stress on the final septum (Fig. 2 and Table 2). The range of septal complexity explored here is limited, the morphology of the Cadoceras sp. septum can be compared to the “Goniatitic” model of Hassan et al. (2002) and the six-wave model of Daniel et al. (1997) as all three models possess only primary flutes, i.e., lobes and saddles (Klug & Hoffmann, 2015).

The fluted septum model of Daniel et al. (1997), with six primary waves, shows elevated tension along the fold axis of the lobes and saddles with compressive stress along the flanks. Hassan et al. (2002) showed elevated tension along the flanks of the flutes in their Goniatitic model with compressive stress concentrated along the fold axis of the lobes and saddles and minor, periodic compressive stress along the periphery of the septum. Our model of Cadoceras sp. shows a very different pattern of tension and compression (Fig. 2). Tension forms along the attachment of the saddles to the shell wall, compression forms in two limited, concave zones between the siphuncular foramen and the suture. The high tension extends along the saddle with increasing amplitude. However we see no notable development of either tension or compression along the flanks nor such a dramatic, clean shift from tension to compression as seen in the models of Hassan et al. (2002) and Daniel et al. (1997). The increase of max. principal stress with increasing amplitude directly contradicts the sutural amplitude index of Batt (1991) who argued that increasing amplitude would improve resistance to hydrostatic pressure by buttressing more of the shell wall following the Buckland model.

All shells demonstrate the concentration of stress along the final septum with, generally, minimal stress developing along the shell wall. This observation is antithetical to the statements of Hewitt & Westermann (1997) and Hassan et al. (2002), that the strength of the septa and the shell wall are very similar if not the same. This result does agree with the explosion experiments and analyses of Kanie et al. (1980) and Kanie & Hattori (1983) where the final septum seems to be the place of initial fracture during implosion and also shows the greatest deformation in stress tests. Progressive loading of N. pompilius (Fig. 4) further demonstrates the accelerated development of stress in the final septum. This being said, the redistribution of stress as parts of the shell break cannot be predicted here; therefore, we cannot comment on the predicted pattern of total failure of the entire shell during implosion.

Septal strength index

We attempted to predict the critical depth for N. pompilius using the septal strength index (Westermann, 1973). There have been other attempts to create a formula to calculate maximum habitat depth, such as the siphuncle strength index (Westermann, 1971) and the index of wall strength (Hewitt & Westermann, 1997). Due to fossil limitations, we do not deal with the siphuncle here and the SSI is a common and simple attempt to estimate habitat depth (Hewitt & Westermann, 1990). The SSI failed to produce consistent results and varied greatly with the choice of region being used to calculate the radius of curvature. Tomographic data allows easy access to a, essentially, infinite number of oblique slices through the septa that can be used to calculate curvature. While one can find a curvature that happens to give a reasonable depth estimate, there is no good criterion with which such a curvature can be chosen reliably between different specimens and morphologies.

The unreliability of the SSI might have motivated the alternative estimates of wall strength mentioned prior. However, the simplistic morphological characterization of the septum, dividing thickness by a radius of curvature, also formed the basis for the current calculation of tensile strength of cephalopod nacre (Westermann & Ward, 1980; Hassan et al., 2002). These calculations also use estimates of the critical hydrostatic pressure which is taken from implosion data. Unfortunately, implosion data for Nautilus has a history of sub-optimal depth control (Westermann & Ward, 1980; Ward, Greenwald & Rougeriye, 1980). Direct measurements of tensile strength were performed but were limited to dry specimens whose organic matter, though they were hydrated, had likely decayed (Currey, 1976).

Point force

The peak max. principal stress measured along the wall decreased when the force was directed along the suture line for all models compared to when the force was directed on the chamber wall (Table 2). However, not all models showed the same magnitude of decrease. Peak tension decreased in N. pompilius by 8%, in S. spirula by 40%, Cadoceras sp. by 41% on the 5th septum, and by 59% on the 10th septum. This result suggests a positive correlation between flute amplitude and max. principal stress due to hydrostatic pressure as max. principal stress increases in the 10th septa compared to the 5th. Conversely, max. principal stress along the suture decreases with increasing flute amplitude. This corresponds with the results of Daniel et al. (1997) who also suggested sutural complexity increases resistance to point loads. Recently, Kerr & Kelley (2015) found no correlation between septal complexity and repair scar frequency and doubted the connection between septal complexity and mechanical resistance to predators. However, it is important to note that shell repair can only occur when damage is done along, or near, the body chamber. Damage to the phragmocone cannot be repaired unless in very rare circumstances and damage to the phragmocone is usually fatal due to drowning (Kröger & Keupp, 2004; Keupp, 2012; Tsujino & Shigeta, 2012). The resistance to point loads supplied by the folded septa would not directly contribute to the strength of the body chamber. Therefore, no correlation between repair scars and septal complexity are to be expected.

Septal complexity and palaeoenvironment

The previously mentioned sutural amplitude index was used by Batt (1991) to assess habitat depth of ammonites of the Cretaceous Western Interior Seaway. Batt (1989) and Batt (1991) studied the change in morphology of ammonite groups during the Western Interior Greenhorn Cyclothem, a transgression-regression cycle, and concluded that increases in sutural amplitude in similar shell morphotypes indicated a deeper habitat. Connections between sutural morphology and environmental changes during transgressions/regressions have been well noted and indicate some connection between morphology and water depth (Bayer & McGhee, 1984; Batt, 1991; Lukeneder, 2015). However, in light of our results, it seems unlikely that this connection with water depth is caused by water depth itself. The maximum depth of the Western Interior Seaway basin is between 250–300 m (Batt, 1989; Batt, 1991). Not only does sutural amplitude weaken the shell to hydrostatic pressure (Fig. 2), even the weakest shell used in this study shows low stress due to hydrostatic pressure at this depth (Fig. 4) and differences in depth on the order of 50 m might not be enough to result in systematic changes in morphology. It seems more likely that changes in water depth results in palaeoecologic changes that cause changes of shell and septal morphology.

Conclusion

This is the first application of FEA to empirical models of cephalopod shells. The limited number of specimens and morphologies explored here mean this is an area ripe for future research. Furthermore, as the final septum seems to be the weakest point of the shell in our models, subsequent FEA of cephalopod shells may not need the entire shell in order to study comparative strength and potential implosion depths. Accurate information on the material properties of cephalopod nacre and the range of these properties within extant cephalopods is needed in order to meaningfully assess the range of implosion depths in fossil cephalopods.

The shell of S. spirula, with a nearly circular whorl cross section and semi-hemispherical septa, shows the highest resistance to hydrostatic pressure while the shell of N. pompilius, which has a more elliptical whorl cross section and dorsoventrally elongated septa, shows the lowest resistance. Due to this, we cannot make a simple connection of septal complexity to hydrostatic pressure. Furthermore, the influence of the overall septal shape—which is limited by the cross-sectional morphology of the aperture—may play a more important role in minimizing the observed max. principal stress when comparing Cadoceras sp. with N. pompilius. Our results suggest a positive correlation of sutural amplitude with max. principal stress due to hydrostatic pressure. However, the septal morphology explored here is limited to primary flutes and higher order flutes may diminish or migrate the principal stress that develops along the saddles.

Cadoceras sp. shows the greatest ability to resist point loads, especially when the load is directed along the suture line. Greater sutural amplitude seems to decrease the max. principal stress developed by a point load along the suture. This suggests that sutural complexity may have helped strengthen the phragmocone against predation.

We would like to thank Julia A Schultz (University of Chicago) for micro-CT scans of Nautilus pompilius. Alexander Lukeneder (Naturhistorisches Museum Wien) for the Spirula spirula specimen from Thailand. We would like to thank the reviewers for their helpful comments.

Additional Information and Declarations

Competing Interests

Author Contributions

Data Availability

The authors declare there are no competing interests.

Robert Lemanis conceived and designed the experiments, performed the experiments, analyzed the data, wrote the paper, prepared figures and/or tables, reviewed drafts of the paper.

Stefan Zachow contributed reagents/materials/analysis tools.

René Hoffmann conceived and designed the experiments, contributed reagents/materials/analysis tools, reviewed drafts of the paper.

The following information was supplied regarding data availability:

Lemanis, Robert (2016): Raw Data for FEA models of the labeled Shells. figshare.

https://dx.doi.org/10.6084/m9.figshare.3475058.v1.

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
