# Peer review of "Comparative cephalopod shell strength and the role of septum morphology on stress distribution"

_PeerJ, doi:10.7717/peerj.2434_

## Round 0.1 · original submission · Major Revisions

We were unable to find a 2nd external Editor, but as an expert in this field, I was able to provide an additional review (see below).

This paper aims at solving a long-standing debate regarding the functional morphology of the mineralized parts of the shells of cephalopods, particularly the outer shell and inner septa. Especially the septa are of evolutionary interest because these can have more or less complex shapes and consist of more or less complex folds. For this purpose, the authors based their work on computer tomographic data of these shell parts obtained from three species, Nautilus pompilius, Spirula spirula, and Cadoceras spec, the latter being an extinct ammonite. In the theoretical model, two cases were specified and analyzed using finite element modeling of stresses induced by either a so-called uni-directional point load versus a more global stress as, for example, induced by hydrostatic pressure. The authors conclude that previous literature is tackling the problem insufficiently, and that previously reported classification systems based on "septal strength index" or materials parameters used for characterizing the building blocks of the shell parts are by far insufficient to elucidate the stress tolerance and maximum depth limits for each species and the mechanical strengths of their respective shells.

The overall approach of this paper is well explained, and there is absolutely no doubt that research was carried out diligently and with great care. However, this manuscript requires a very deep background in the field of cephalopod shell morpholgy, functional complexity, evolution and ecology as well as a fundamental understanding of mechanics of materials and computational materials science. The manuscript lacks clarity in terms of why the three species were chosen, and how exactly the three selected shells were analyzed. It also remains unclear in which regards the specimens differed (e.g. in size, developmental stage, preservation over time, etc.) and how these differences would influence the outcome of the CT data that went into the model. The literature is well referenced, but without studying several of the cited papers, the present work is extremely difficult to understand. In principle, the data are concisely reported. However, due to the extremely interdisciplinary nature of this work it will help a lot to put more effort in presenting and discussing each experiment much more clearly. Otherwise, it remains a matter of speculation how the conclusions could have been reached.

Recommendations for improvement:

- Improve the background and context by including a scheme with the three shells and septa, their morphology as relevant to the present findings, and respective scale bars.
- Explain the figures and data in more detail to make it easier for non-experts in the field to understand their meaning
- Supply raw data were applicable
- Explain more clearly in which respect each shell part can be comparatively analyzed across species.
- The complex morphology of septa seems to pose several problems in terms of mechanical stability, for example it remains unclear how homogenous these materials are. For example, Klug & Hoffmann (2015) as cited in the present work describe septa with canals (e.g. Fig. 3.5 in the genus Gaudryceras). Also, the reference calcite (as mentioned several times in cited literature), a very brittle material, does not seem appropriate for comparing it with nano-composite materials such as cephalopod nacre (see e.g. Ashby's textbook on nanomaterials and/or biological materials). At least for non-expert readers, it will be difficult to understand why it seems acceptable here to assume an average Young's modulus of 50 GPa for each model, not taking species-specific substructures and anisotropies of nacre-type materials into account. Would other control experiments with nanocomposite structures be possible and, if not, why (e.g. how long does such an FEA take?).
- Results section: Clear and unambiguous symbols should be used throughout the data sets.

editorial criteria summary:

BASIC REPORTING
YES Clear, unambiguous, professional English language used throughout.
NO Intro & background to show context.
YES Literature well referenced & relevant.
YES Structure conforms to PeerJ standard, discipline norm, or improved for clarity.
NO Figures are relevant, high quality, well labelled & described.
NO Raw data supplied (See PeerJ policy).

EXPERIMENTAL DESIGN
YES Original primary research within Scope of the journal.
YES Research question well defined, relevant & meaningful.
YES It is stated how research fills an identified knowledge gap.
NO Rigorous investigation performed to a high technical & ethical standard.
NO Methods described with sufficient detail &information to replicate.

VALIDITY OF THE FINDINGS
YES Impact and novelty not assessed.
? Negative/inconclusive results accepted.
NO Meaningful replication encouraged where rationale & benefit to literature is clearly stated.
? Data is robust, statistically sound, &controlled.
NO Conclusion well stated, linked to original research question & limited to supporting results.
YES Speculation is welcome, but should be identified as such.

·

Basic reporting

Article is clear

Experimental design

Experimental design is strong

Validity of the findings

Findings are clear and appropriate.

Additional comments

The paper describes a methodology to both image and subsequently model ammonoid mechanical properties under hydrostatic pressure. The effect of the folded septa on mechanical performance is a particular focus. The methodology is extremely powerful and I recommend publication of the work based on this. Perhaps the weakest part of the paper is the outcomes of the work, as a lack of more general rules is found, but the work is still a rigorous contribution. A number of improvements are needed, listed below, but I am happy to recommend publication after the authors have addressed each point.

Line 120. Correct the equipment name.
Line 121-126. Perhaps further discussion of the voxel size is needed. I expect that the voxel size here is defined by the field of view but discussion might be required to identify the structural features missing when the voxel size is high.
Line 144-148. The authors indicate that the original (relatively large) dataset and downsampling by 2x give similar FE model results. The authors should make clear the FEA analysis carried out to show that the results are similar. The authors should also indicate the changes for the 4x and 6x downsampling i.e. when is a result ‘similar’ and when are clear differences observed due to the downsampling.
Line 166-173. To be clear, the FEA assumes the material behavior is linear elastic and of infinite strength?
Line 191. I am unclear on the point force scaling here. Why apply a scaling as the models are considering structures of different sizes that are presumably able to resist similar point forces?
Line 195. Define the strength index as this is presumably different to strength.
Line 208. Unclear. First simply state what Fig 1 is showing and then make comments.
Table 2. Correct the units in the first row.
Table 2. A significant amount of data is shown but are the authors both confident in the accuracy of these numbers (to two decimal places) and the possible error?
Line 283. Where is this shown?

---

## Round 0.2 · accepted · Accept

The revised manuscript addresses all major points raised during the reviewing process.

·

Basic reporting

Revision addresses my previous points

Experimental design

Revision addresses my previous points

Validity of the findings

Revision addresses my previous points

Additional comments

Revision addresses my previous points